# Consumption-Driven Carbon Emission Reduction Path and Simulation Research in Steel Industry: A Case Study of China

Desheng Xu [1,2], Encui Liu [1,2], Wei Duan [1,2,*] and Ke Yang [1,2]

[1]  School of Economics and Management, Inner Mongolia University of Technology, Hohhot 010051, China
[2]  Inner Mongolia Management Modernization Research Center, Hohhot 010051, China
*   Correspondence: duanwei1246@imut.edu.cn

**Abstract:** China's steel industry's carbon emissions accounted for more than 60% of global carbon emissions, approximately 15% in China in 2020. China's steel industry accounted for approximately 16% of China's total carbon emissions in 2021. The ability to reduce the carbon dioxide emissions generated by the steel industry and protect the living environment for humans and nature has become a realistic issue for China. This paper constructs a steel consumption–carbon emission system. Research shows that by adjusting the GDP growth rate and $CO_2$ emissions per unit of steel production, the carbon peak in the steel industry will advance to 2030 and the carbon emissions after the peak will be significantly reduced. The reduction in steel consumption in the construction and machinery sectors does not have a significant impact on carbon emissions from the steel industry, whereas the reduction in steel consumption in the transportation and infrastructure sectors has contributed to carbon reduction activities in the steel industry. When all four sectors are regulated simultaneously, it is found that the predicted carbon peaking time for the steel sector advances to 2029, fulfilling the goal of achieving carbon peaking by 2030. Carbon emissions should decrease after that point.

**Keywords:** steel consumption; carbon emissions; system dynamics; carbon peak

## 1. Introduction

The steel industry, one of the major energy consuming industries in China, was severely impacted by the increase in global warming and environmental degradation. The government and academia have started to reflect on the related carbon emission issues. Reducing carbon dioxide emissions from the steel industry and protecting the living environment of human beings and nature have become a real issue for China. The green and healthy development of the steel industry plays an important role in protecting China's ecological and social environment. It is important to reduce China's carbon emissions and promote sustainable economic and social development. The study of carbon emissions in China's iron and steel industry from the perspective of consumption is essential. This is beneficial to the formulation and implementation of future carbon emission reduction policies in China's steel industry.

Carbon dioxide produced by human activities ($CO_2$) has become the leading cause of global warming and environmental degradation [1]. Industry accounted for 24% of global $CO_2$ emissions in 2020, and steel is one of China's energy-intensive industries, accounting for approximately 33.8% of total industrial $CO_2$ emissions [2]. China's steel industry emitted nearly 1.8 billion t of carbon, accounting for approximately 16% of the country's total carbon emissions, in 2021 [3]. According to the World Iron and Steel Association, China remained the world's largest steel producer in 2020, with its output reaching 1.065 billion tons, accounting for 56.71% of the world's total capacity, up 6.97% from a year earlier [4]. That same year, China's steel industry accounted for more than 60% of worldwide carbon emissions and 15% of the country's total [5].

In 2020, China's GDP totaled 14.72 trillion USA dollars [6], the second largest economy in the world. China's carbon emissions continue to grow, along with its rapid economic

development. By 2020, China's total carbon dioxide emissions reached 9.899 billion tons, accounting for 30.7% of global emissions [7]. In September of the same year, Chinese President Jinping Xi announced at the 75th UN Conference that China would strive to achieve carbon peak by 2030 and carbon neutrality by 2060 [8,9]. Carbon peaking and carbon neutral green low-carbon development strategies are constantly receiving attention from all sectors of society. As one of the main sources of carbon dioxide emissions, the steel industry is crucial to the implementation of China's low-carbon development strategy to achieve its carbon peak and carbon neutral goals. No matter from the industry level or from the local level, the steel industry is the key area to implement the "double carbon" target. It is also a key industry that needs to take the lead in achieving the carbon peak. The achievement of its carbon peak and carbon neutral targets is crucial to the implementation of China's low carbon development strategy. Therefore, through the analysis of steel consumption influence on China's steel industry carbon emissions, the carbon steel consumption system dynamics model was constructed. For the Chinese steel industry, finding the future development trend for carbon emissions and the carbon peak simulation forecast, as well as promoting China's carbon peak and achieving the goal of carbon neutrality, have a certain theoretical and practical significance.

The results show that the industries consuming steel can accelerate depreciation in a short period of time and shorten the service life of steel products and equipment. In the long term, it can improve science and technology, implement energy substitution and lightweight development, actively develop emission reduction pathways, and actively build low-carbon infrastructure and green transportation. It can reduce steel consumption and reduce the carbon emissions of the steel industry so that the steel industry's carbon peak time advances to 2029, smoothly achieving the carbon peak target.

## 2. Literature Review

### 2.1. Steel Consumption Research

Forecasting steel consumption and demand is important for the development of steel and its related industries in China. The general domestic and international social environment requires China to reduce the pressure on carbon emissions. These require that we make more accurate forecasts concerning steel consumption demand and that the corresponding scenario analysis is carried out based on the obtained forecast results.

Wang, Li (2012) predicts steel consumption demand based on a time series model, and then combines it with grey system, scenario-based multiple regression model forecasts and grey system models for mixed forecasts, with corrections to account for cyclical factors. Finally, the forecast is combined with expert forecasts to obtain the steel consumption forecast. Such forecasts take into account both the objectivity and the full verifiability of the model forecasts [10]. Guan Gong (2014) established a support vector regression machine to study the Chinese steel consumption demand forecasting model. He found that the support vector regression machine has the characteristics of high prediction accuracy and strong interpretation in solving small sample and nonlinear problems. Its prediction accuracy is higher than that of the BP neural network and GM model, and it is a better method and tool for forecasting steel consumption [11].

From an overall macroeconomic perspective, Xuehong Zhu and Anqi Zeng et al. (2019) constructed a recursive dynamic steel industry chain CGE model to explore the impacts received by the macroeconomy, market and technology from a chain perspective under three different levels of environmental regulation in the steel industry. The simulation results show that the real GDP and output of the downstream sectors of the steel industry will be slightly affected. In the long run, the output and exports of the steel industry itself would benefit from strict environmental regulation [12]. Fan Fengyan and Du Qingkun (2017) first explored the response relationship between energy consumption and economic growth in the steel industry in China from 1995–2014 using an elastic decoupling model. Then, the Fisher decomposition method was applied to decompose the changes in energy consumption of the steel industry. The results show that economic growth has a pulling

effect on energy consumption in the steel industry, with a cumulative effect of 11.0543. Energy intensity has a suppressive effect on energy consumption in the steel industry, with a cumulative effect of 0.3385. To achieve absolute decoupling of economic growth and energy consumption in the steel industry, technological progress should be promoted to further reduce energy consumption intensity. At the same time, the capacity structure should be optimized and adjusted to improve the energy use efficiency of the steel industry [13].

From the perspective of demand structure, Yu Yihong and Zhang Huaxiang (2013) distinguish steel into two categories, consumption-pull and investment-pull, according to the different uses of steel. The correspondence between investment activities and domestic consumption and steel downstream industries is sorted out. The long-run equilibrium relationship between consumption-driven steel and automobile and household appliance production is verified through empirical analysis. Based on the analysis of the structural characteristics of steel demand and its relationship with economic transformation, we make a judgment on the future development trend of China's steel industry in the light of the structural changes in steel demand. Suggestions are also made for government industrial policies and manufacturers' strategies [14]. Gu Xiaowei, Qin Zongchen, Wang Qing and Wang Fengbo (2017) developed an incremental decomposition model of iron ore consumption intensity. Applying the data in the input–output table of national economy, the incremental decomposition of iron ore consumption intensity was calculated and analyzed by time periods. It was found that the main contradiction that needs to be solved in the iron and steel smelting and processing industry is to eliminate the excess low-end production capacity and promote the upgrading of product structure, reducing the iron ore consumption intensity of the whole economy to achieve a healthy and stable development of the industry [15].

The increase or decrease in steel consumption has a crucial impact on related environmental issues, especially in terms of carbon emission reduction. Based on the scenario analysis, Wang, X.Y. et al. (2022) found that in the long term, the crude steel consumption demand is still relatively strong, and the crude steel production is at the peak plateau in the 14th Five-Year Plan. Further strengthening the carbon emission control based on the existing efforts can promote the carbon emission peak of China's steel industry one to three years earlier. If the current crude steel consumption demand is close to the peak and will gradually decrease in the future, strengthening the carbon emission control can push the carbon emission peak of China's steel industry 0–1 years earlier [16]. Wang Qianfeng and Xia Dan (2020) innovatively proposed the "coal consumption reduction elasticity coefficient" to evaluate the direct effect of coal consumption reduction in each industry on reducing various industrial pollution. Furthermore, with the help of a panel regression model, we analyzed the key factors affecting the coordination of industrial coal consumption and pollution emission in China at this stage. The panel regression results confirm that coal consumption reduction has a significant direct effect on industrial pollution reduction. It also identifies that R&D innovation and pollution control investment in each industry are important factors for further coordination between energy consumption and environmental pollution, which has important implications for the future goal of achieving efficient resource use and coordinated development with the environment in Chinese industrial industries [17]. Lubomír Klimes, Michal Brezina and Tomas Mauder et al. (2020) argue that the steel industry represents a series of technological processes. These processes consume large amounts of energy and water and generate large amounts of emissions. To study the consumption of water, the first thing that needs to be analyzed is the consumption of steel, and the requirements in the analysis of processes in the steel industry are very strict in determining the energy, water and emission processes [18]. The high energy consumption and the large amount of pollutants emitted by high energy consumption industries during the production process pose a great threat to the ecological environment. Li and Yang (2017) base their research on the basic concepts of circular economy theory and sustainable development to study energy saving, emission reduction and ecological economy construction in high energy consumption industries. They found that the reduction in steel consumption

contributes significantly to the reduction in total energy consumption. In recent years, the energy consumption per ton of steel has been decreasing year by year, and energy saving and emission reduction measures such as technological reform, policy regulation and control and economic coordination have played a major role. The ecological economic system of energy conservation and emission reduction based on circular economy can alleviate the problem of high energy consumption in the industry and lay the foundation for the ecological development of the industry [19].

To sum up, it is a trend to study and analyze carbon emissions in the steel industry from the perspective of consumption. Iron and steel industry is the pillar industry in China. The demand for economic development has led to a large amount of $CO_2$ emissions causing environmental pollution. Therefore, a scientific forecast of the long-term demand for domestic steel is not only an important reference value for the layout of China's steel industry and the configuration of the global industry chain but also beneficial to the reduction in carbon emissions in the steel industry. It provides an important basis for strategic decisions for China's economic development.

### 2.2. Carbon Emission Research in the Steel Industry

Under the background of carbon peak and carbon neutrality, green development decision-making has been paid more attention by the country and society, and the carbon emission of the steel industry has attracted more and more scholars' attention. Existing studies on carbon emissions from steel industry can be generally divided into four categories:

The first is the analysis of the econometric model. From a macro perspective, Wen Lei and Shao Hengyang (2019) used a data-driven non-parametric additive regression model to investigate the main influencing factors of $CO_2$ emissions in China. They found that the non-linear effect of the economy on $CO_2$ emissions is consistent with the environmental Kuznets curve hypothesis, and industrialization also exhibits a subversive "U-shaped" relationship. As China's economy is developing at a high speed, the country has a large demand for steel, cement and other raw materials, which leads to more carbon dioxide emissions [20]. Ya Chen et al. (2020) used an econometric model and found a significant inverted U-shaped relationship between environmental regulations and $CO_2$ emissions in the steel industry. In addition, there is regional heterogeneity and regulatory intensity in the effect of environmental regulations on $CO_2$ emissions. High environmental regulations have an inverted U-shape impact on $CO_2$ emissions, while low environmental law rules show a U-shape. The mechanisms by which environmental regulations affect $CO_2$ emissions are synergistic and technological innovation effects [21]. From a micro perspective, Wei Li and Chen Weida et al. (2021) constructed a production cycle model with profit maximization as the goal. They found that steel companies can adjust their blast furnace-converter and electric furnace steelmaking processes to meet market demand and achieve efficient resource allocation. If the market demand signal is high, the two steelmaking methods are used at the same time; otherwise, only blast-converter steelmaking is used. Under the current trend of environmental transformation, the research results provide certain theoretical and strategic supports for iron and steel enterprises to actively promote green steel production [22]. Sensen Zhang et al. (2022) used the energy balance sheet splitting method, STIRPAT model, gray correlation method and GA-BP neural network model to study and predict the carbon emissions of the construction industry in Jiangsu province. The research results show that steel production has a significant catalytic effect on carbon emissions. Therefore, in the process of carbon emission reduction in the construction industry in Jiangsu province, it is important to reduce the use of steel as much as possible and switch to new materials that are energy efficient and reduce emissions as an alternative [23].

The second is bottom-up analysis. The steel industry is one of the largest energy consumption and carbon emission industries in China, which is of great significance to China's overall low-carbon transformation development path. Liu Junling et al. (2019)

found by constructing a "bottom-up" energy system model including industrial modules: First, under the conditions of full utilization of emission reduction potential, carbon emissions in industrial sectors, including the steel industry, are expected to be reduced by nearly 4 billion tons by 2050. Second, we need to improve energy efficiency in a timely manner. After 2030, we need to vigorously promote CCS technology and embark on a transformation path of deep decarbonization, which is also an important driving force for reducing carbon emissions [24]. In addition, carbon emissions from the steel industry have a significant impact on climate change in recent years. Nidheesh and Kumar (2019) believe that the large-scale production of cement and steel plays a critical role in the development of a country. However, their mass production has led to serious environmental degradation. Therefore, sustainable production policies should be vigorously implemented in these areas. For example, carbon sequestration can effectively reduce carbon dioxide emissions in the steel industry, and the by-products of steel production can be used as raw materials for paint, cement, fertilizer and so on [25]. Du Guang, Sun Chuanwang and Ouyang Xiaoling et al. (2018) used a top-down analysis method to study and found that the increasing industrial scale and the weakening energy intensity are the key reasons for the increase and decrease in $CO_2$ emissions from energy-intensive industries. Moreover, the impact of energy intensity is particularly significant in the petroleum coking industry, steel industry and power industry, which is a necessary factor to be considered to curb global warming and protect the environment [26].

The third is process analysis. Lubomir, Michal and Tomas et al. (2020) argue that the steel industry represents a series of technological processes that not only consume a large amount of energy and water but also produce a large amount of waste gas, etc. A large number of pollutants produced in the process of its production have caused a very serious impact on the ecological environment [18]. Li and Yang (2017) found that the reduction in steel consumption contributed greatly to the reduction in total energy consumption. Energy-saving and emission-reduction measures such as technological reform, policy regulation and economic coordination play a major role. The energy-saving and emission-reduction ecological economic system based on circular economy can alleviate the problem of high industrial energy consumption and lay a foundation for the ecological development of the industry [19]. Although the carbon emission intensity of China's steel industry has decreased after years of governance, the demand for steel and production scale continues to increase under the rapid development of China's economy. Wang Yafei (2018) used the process analysis method and found that combining the advanced experience of developed countries and the existing problems of carbon emissions in China's steel industry is necessary. China should integrate production capacity, maintain long-term and stable high-level environmental research and development strength, and pay attention to the breakthrough developments in energy-saving technology [27]. In addition, Zhang Fujun et al. (2022) concluded that the Chinese steel industry is in a state of "carbon lock-in". Only through simultaneous technological and institutional changes can "carbon unlocking" be achieved. The "carbon unlocking" can be achieved through reasonable limitation of steel production capacity and increasing the proportion of all-scrap electric arc furnaces, as well as through the development and application of hydrogen metallurgy and carbon capture, utilization and storage technologies and institutional changes [28].

The fourth is situational analysis. Process structure transformation is the key "dividend" of energy saving and emission reduction in China's steel industry in the future. In the later carbon reduction process, energy saving technology will play a more and more obvious role. To achieve the goal of cutting emissions by half by 2050, China's steel industry needs to implement corresponding measures, such as capacity restructuring, scrap recycling and promotion of cleaner production technologies, over the next three decades [29]. Through scenario analysis, Li Xinchuang and Li Bing (2019) concluded that the low-carbon transformation development path of China's steel industry mainly includes three directions: demand reduction, energy efficiency improvement and technological innovation. Among them, reducing demand is mainly reflected in reducing raw fuel

consumption, optimizing process structure, reducing steel consumption and promoting the coupling development of circular economy. The improvement of energy efficiency is reflected in the transformation of energy-saving and low-carbon technologies and the improvement of the application of non-fossil energy. The key point of process innovation is to use non-carbonization technology to achieve large-scale carbon reduction and decarbonization [30]. In 2020, China announced that $CO_2$ emissions are set to peak by 2030. Steel industry is a typical resource—and energy-intensive industry, and it is also a key industrial sector that needs to implement the carbon peak target as soon as possible. Promoting inter-industry coupling and improving carbon asset management are of great significance to the low-carbon transformation of China's steel industry [31]. Songyan Ren et al. (2022) developed the ICEEH-GD model and designed an energy security scenario (ES), a restricted high carbon emission industry scenario (RHS) and an integrated policy scenario (CP) to study the impact of policies to restrict high emission industries and renewable energy on investment structure, macroeconomic and social transformation in Guangdong Province. The study concluded that the power sector should increase investment in renewable energy to ensure energy security; limit new capacity in high-emission industries such as cement, iron and steel; and increase green transformation and efficiency improvement in existing high-emission industries [32].

### 2.3. Application of System Dynamics

System dynamics, as a system thinking and system modeling tool, has strong applicability and flexibility. Taking into full consideration the complexity and dynamics of the system, it is widely used in the evaluation studies of the effects of various industrial policies.

In the context of carbon emission reduction and power market reform, the market-based carbon emission reduction policy of the power generation industry has a significant impact on the process of power market reform and carbon emission reduction effect. To this end, Jinliang Zhang and Xiuxiu Zhou (2020) studied the market-based carbon emission reduction policy of the power generation industry. They also constructed a system dynamics model based on the green certificate trading market, carbon emissions trading market, power generation rights trading market and electricity market among them. The impact on carbon emission reduction and the electricity market under different policy scenarios was analyzed [33]. Zhan-Yong Wang, Min Liu and Wen-Bin Xiong (2020) took China's nuclear power industry as a research object. By sorting out the industry chain structure and data, a system dynamics simulation was conducted by drawing on the existing research base. The competitive advantages and challenges of China's nuclear power industry chain in domestic and international power markets were analyzed. They combined the existing industrial policies for the safe and efficient development of nuclear power to propose policy recommendations for the sustainable development of China's nuclear power industry [34].

In the energy conservation and environmental protection industry, Zhou Xin, Zhen Hong and Zhao Nan (2020) used system dynamics principles and methods to construct a system dynamics model of port energy conservation and emission reduction, dynamically simulated the implementation effects under different policy scenarios and conducted a comparative analysis. In turn, they put forward opinions and suggestions for the development of energy conservation and environmental protection industry in China [35]. Hao Weibao, Lei Yanxi and Li Shengguo et al. (2019) used the system dynamics approach to construct a system dynamics model for the development of energy conservation and environmental protection industry. Policy simulations were conducted from four perspectives: current policy status of energy conservation and environmental protection industry development, industrial policy program, fiscal policy program and technology policy program. It was found that adjusting industrial policies can effectively promote the development of energy and environmental protection industries in China [36]. Sungwook Yoon and Sukjae Jeong (2016) analyzed and evaluated the economic and environmental benefits of the international

aviation industry in Korea under different policies through a system dynamics model to explore better policies for greenhouse gas emission reduction.

As for the sustainable development of the building industry, Liu Jing and Zhao Jingyun (2018) analyzed the influencing factors of building carbon emissions based on the system dynamics model. Different simulation scenarios were set to predict the future development trend of building carbon emissions and its internal logical relationship. In turn, it provided a theoretical basis and emission reduction paths for reducing building energy consumption, reducing total carbon emissions and achieving the green and sustainable development of buildings [37]. Yao et al. used a system dynamics approach to analyze the technical and economic feasibility of carbon capture from the perspective of the steel supply chain, taking steel enterprises as the research object. It was found that under specific conditions, steel companies can successfully apply carbon capture technology. An appropriate government subsidy program was established to foster environmental awareness among downstream consumers to increase the willingness to purchase low-carbon steel [38].

Existing studies have conducted a series of investigations and analyses of future development scenarios and related policies in various industries using the system dynamics approach and have achieved certain results. At the same time, system dynamics, as a system thinking and system modeling tool, has strong applicability and flexibility. It can be competent for scenario analysis problems in various industries. Therefore, using the system dynamics approach from a systematic and holistic perspective, this simulation modeling method is applied to the research and analysis of carbon emission problems in the steel industry based on consumption-driven scenarios. It is feasible for grasping the carbon emission reduction policies and measures that should be implemented in the future from a holistic perspective.

### 2.4. Contribution of This Article

In summary, from the perspective of implementing measures and subjects to reduce emissions in the iron and steel industry, scholars from all walks of life have conducted studies on carbon emissions from the perspective of production in the iron and steel industry. Low carbon emission reduction is mainly carried out from the production side. There is a relative lack of research on the consumption side of the iron and steel industry. Therefore, this paper summarizes the previous studies on carbon emissions in the iron and steel industry and uses a system dynamics approach to build a scenario interaction model to analyze the consumption side of the iron and steel industry and its carbon dioxide emissions from a system perspective. Thus, it provides a useful reference for the formulation of carbon emission reduction policies and the implementation of sustainable development strategies in China's steel industry.

### 3. Dynamic Model Construction of Steel Consumption-Carbon Emission System

### 3.1. Modeling Steps for System Dynamics

The main process of system dynamics modeling is shown below.

(i) Overall analysis: In this stage, the main focus lies on analyzing the problem in depth and identifying the main factors.

(ii) Structural analysis: This stage mainly deals with the various information present in the system and analyzes the feedback mechanism of the system.

(iii) Modeling: (1) establishing equations, expressing state variables, rate variables and auxiliary variables in mathematical form; (2) determining and estimating parameters; (3) assigning table functions to values.

(iv) Run the test: Using Vensim DSS 6.1C, a simulation software for system dynamics, construct the model and run the model.

(v) Testing and evaluation: After running the model, in order to make sure that the model can reflect the real system as much as possible, it is necessary to test the model by adjusting and modifying the relevant parameters so that the model can reflect the

real system as much as possible. It is mainly divided into the validity test and the sensitivity test.

(vi) Simulation and prediction: input a variety of pre-designed scenarios into the simulation model, obtain the trend results desired for analysis, and analyze and discuss the trend results obtained to provide a strong basis for decision-making and regulation.

### 3.2. System Variables and Flow Diagram

The purpose of constructing a simulation model of the steel consumption–carbon emission system is: Steel consumption is mainly analysis—carbon interaction relationship between each variable in the system, simulation of the Chinese steel industry in 2009–2050 carbon steel consumption—system dynamics, and discovery of the main policy factors affecting the development of the system, as shown in Figure 1. Through the system simulation and prediction, analysis of the steel industry carbon emissions, a different policy combination plan can be set. Analysis of the peak time of carbon emissions in steel industry under different scenarios can ensure that the "30 60 target" is completed on time or even in advance.

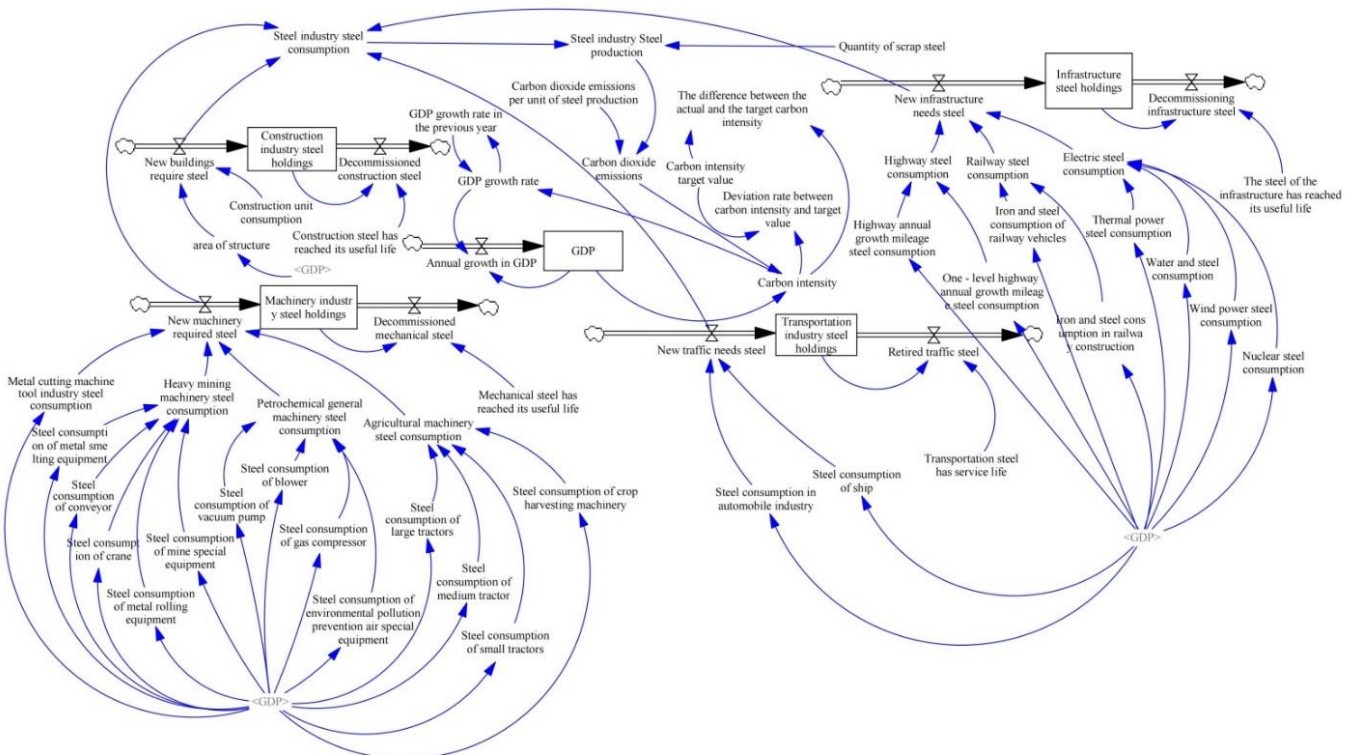

**Figure 1.** Flow diagram of the steel consumption–carbon emission system.

### 3.3. Determining System Parameters

The construction industry, machinery industry, transportation industry and infrastructure industry in the steel consumption–carbon emission system model are determined from the conclusion report of the Simulation Study on Carbon Reduction Mechanisms in China's Steel Industry supported by the National Natural Science Foundation of China. In addition to the above four big industries, the report involved the light industry and other industries. Since the steel consumption of these industries is relatively small, they have little impact on the carbon emissions of the steel industry. This paper mainly lists the four major industries of construction, machinery, transportation and infrastructure to analyze the impact of steel consumption on the carbon emissions of the steel industry.

In order to ensure the accuracy of the data used, in view of the annual data that can be collected, the system historical data were selected from 2009 to 2018, a total of 10 years, with 2009 as the base period of simulation. The original data sources are China Industrial Statistical Yearbook, China Machinery Industry Yearbook, China Transportation Statistical Yearbook, China Statistical Yearbook, etc.

In the carbon emission system model based on steel consumption in various industries, the following variable parameters are usually used: GDP, steel storage in various industries, steel consumption in the steel industry, $CO_2$ emissions, etc. Due to space constraints, this paper lists only a few major quantitative equations related to carbon dioxide emissions and regulatory factors.

(1)  $CO_2$ Emissions = unit of $CO_2$ emissions of steel production * Steel production of steel industry
(2)  Carbon intensity = $CO_2$ emissions/GDP
(3)  Deviation rate between carbon emission intensity and target value = (carbon emission intensity—target value of carbon emission intensity)/target value of carbon emission intensity
(4)  GDP growth rate of the previous year = DELAY FIXED(GDP growth rate, 1, 0.06)
(5)  GDP growth rate = GDP growth rate of the previous year * IF THEN ELSE(Carbon emission intensity $\leq$ 0, 1, 0.95)
(6)  Steel industry steel production = steel industry steel consumption + scrap quantity

The parameters of the steel consumption–carbon emission system can be determined by means of average value, direct assignment and regression analysis. For the constants with small numerical changes that were missing in a few years, the average value method was adopted to assign values to various variables based on historical data, such as the industrial steel consumption of metal cutting machine tools in 2012, the steel consumption of gas compressors in 2012 and the steel consumption of blowers in 2013. For the initial values of exogenous variables, a simple direct assignment method was used to sort out and calculate them according to the historical data of statistical yearbooks, such as GDP growth rate, building unit consumption and scrap quantity. For variables with missing data in certain years and in line with univariate or multivariate linear regression, values were assigned according to regression equation calculation, such as steel consumption of large, medium and small tractors and crop harvesting machinery in 2012.

## 4. Validity Test of the System Model

System dynamics modeling is mainly to analyze and solve the problem systematically, and the effectiveness of the system directly affects the operability of the model. To verify that the model is correct and reasonable, a historical check must be performed.

By using Vensim DSS 6.1C software, the validity of the model was verified according to the flow diagram and parameter setting of the carbon emission system. The simulation interval was set as 2009–2018, and the simulation step was set as 1 year. In order to determine the consistency between the model running state and the actual situation, it was necessary to verify the validity of the built model. If the deviation between the actual running results and the actual data is small, the model design is reasonable and feasible, and can reflect the development trend of the system. Finally, each parameter was simulated and verified by using the established model. This paper only lists the carbon emission indexes that need to be regulated and the GDP indexes that have an impact on the development of steel consumption. Table 1 shows the GDP simulation value, real value and error rate of this model.

**Table 1.** Simulation results and error rate of GDP.

| Time/Year | Real GDP/1000 Billion RMB | GDP Simulation Value/1000 Billion RMB | Error Rate/% |
|---|---|---|---|
| 2009 | 34.8518 | 34.8518 | 0.00 |
| 2010 | 41.2119 | 38.9663 | 5.45 |
| 2011 | 48.7940 | 43.5665 | 10.71 |
| 2012 | 53.8580 | 48.7099 | 9.56 |
| 2013 | 59.2963 | 54.4604 | 8.16 |
| 2014 | 64.3563 | 60.8898 | 5.39 |
| 2015 | 68.8858 | 68.0783 | 1.17 |
| 2016 | 74.6395 | 76.1154 | 1.98 |
| 2017 | 83.2036 | 85.1014 | 2.28 |
| 2018 | 91.9281 | 95.1482 | 3.50 |

As can be seen from Table 1, the large absolute error rates between the simulated value and the real value of variable GDP in the model are 10.71% (2011), 9.56% (2012) and 8.16% (2013). The absolute error rates in other years except these three years are all controlled within 6%, and the average absolute error is 3.27%, i.e., less than 4%. It can be seen that the error between the system simulation results and the actual results is small and the fitting degree is high, which meets the error requirements of the complex system model simulation. The model running status and system parameters are set reasonably, which can effectively grasp the variation law and correlation between variables in the steel consumption–carbon emission system. The model is used to simulate and forecast the change in trend of carbon emission of steel industry in the future.

## 5. Simulation of Carbon Emissions from Steel Industry under Different Scenarios

In this paper, seven different development scenarios were set up to adjust the system parameter values of GDP growth rate, construction industry, machinery industry, transportation industry and infrastructure industry, and analyze and forecast the future impact of carbon emissions in steel industry under different development scenarios.

### 5.1. System Simulation Prediction under Baseline Scenario (Scenario 1)

This scenario was mainly to simulate and forecast the development and change in trend of carbon emissions of steel industry from 2019 to 2050, and the simulation step was set as one year.

As can be seen from Figure 2, according to the current system behavior, the carbon emissions of the iron and steel industry from 2019 to 2050 were simulated, presenting an inverted "U" shaped trend. Among them, the growth rate was relatively fast from 2019 to 2031 and relatively gentle from 2031 to 2035; the peak appeared in 2033 and began to show a slow decline in trend from 2033 to 2050. In 2033, China's steel industry will achieve 200.97 billion tons of carbon dioxide emissions with an average annual growth rate of 0.83%. Figure 3 shows that the GDP will increase rapidly from 2019 to 2050, reaching 32,330 trillion yuan by 2050, more than nine times that of 2009. From 2019 to 2050, the carbon emission intensity of China's steel industry will maintain a stable downward trend. It follows that in the baseline scenario, the predicted time of carbon peaking in China is 2033, and GDP is increasing. This may indicate that with the high GDP growth, China will achieve a technological and institutional change in carbon emission reduction technology as found in the previous literature after 2033.

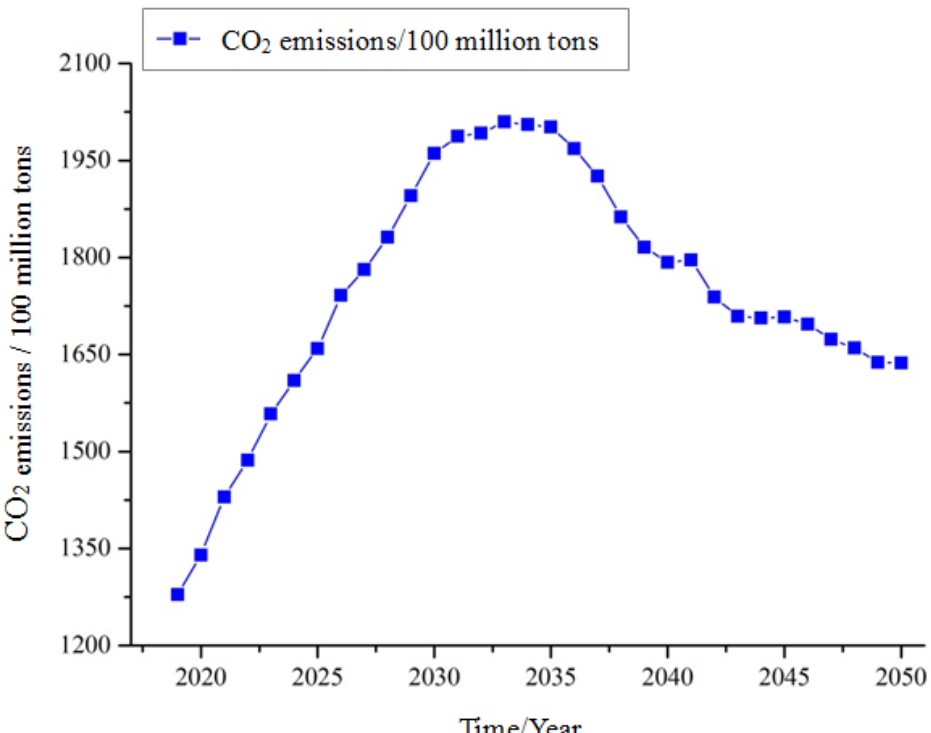

**Figure 2.** Simulation forecast of $CO_2$ emissions in the steel industry from 2019 to 2050.

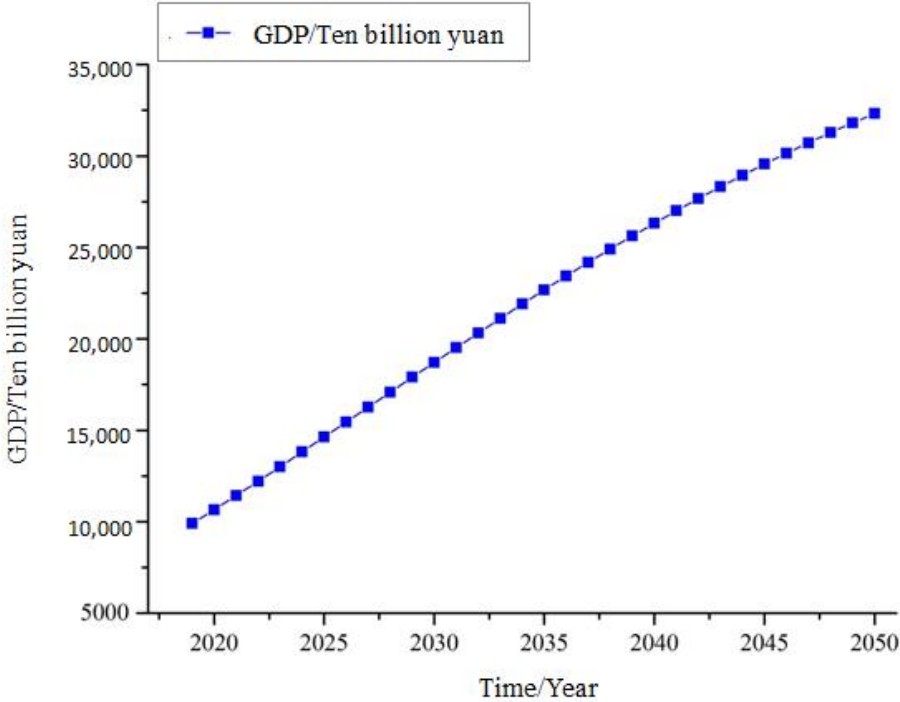

**Figure 3.** GDP simulation prediction from 2019 to 2050.

With regard to achieving the goal of low-carbon development of global steel, the IEA (International Energy Agency, Paris, France) believes that through strict control of steel demand and comprehensive application of various low-carbon technologies, direct carbon emissions in 2050 will be reduced by 50% compared with 2020, and carbon emissions per ton of steel will be reduced by more than 50%. As shown in Table 2, the carbon emission intensity of the steel industry in 2050 will be reduced by 60.76% compared with 2019

and 59.72% compared with 2020, meeting the requirements of low-carbon development policies. The policy simulation of different scenarios can be continued so as to provide some suggestions for reducing carbon emissions in the steel industry to reach the peak target.

**Table 2.** Simulation and prediction data results of carbon emission intensity of steel industry based on the baseline scenario from 2019 to 2050.

| Time/Year | Carbon Emission Intensity/Million Tons/Billion Yuan | Time/Year | Carbon Emission Intensity/Million Tons/Billion Yuan |
| --- | --- | --- | --- |
| 2019 | 1.29034 | 2035 | 0.88269 |
| 2020 | 1.25688 | 2036 | 0.83995 |
| 2021 | 1.25080 | 2037 | 0.79643 |
| 2022 | 1.21679 | 2038 | 0.74780 |
| 2023 | 1.19741 | 2039 | 0.70867 |
| 2024 | 1.16508 | 2040 | 0.68093 |
| 2025 | 1.13425 | 2041 | 0.66508 |
| 2026 | 1.12738 | 2042 | 0.62834 |
| 2027 | 1.09505 | 2043 | 0.60360 |
| 2028 | 1.07177 | 2044 | 0.58954 |
| 2029 | 1.05869 | 2045 | 0.57794 |
| 2030 | 1.04764 | 2046 | 0.56287 |
| 2031 | 1.01774 | 2047 | 0.54482 |
| 2032 | 0.97995 | 2048 | 0.53074 |
| 2033 | 0.95137 | 2049 | 0.51490 |
| 2034 | 0.91533 | 2050 | 0.50633 |

According to China's emission reduction target, the carbon emission intensity of the steel industry was reduced to 502.54 million tons of carbon per billion yuan by 2020, more than half of the 2005 carbon emission intensity (102.99 million tons of carbon per billion yuan, constant price in 2003). Therefore, based on the development law of the current system behavior, it is predicted that China's carbon emission intensity reduction target of 45% compared with 2005 in 2020 can be exceeded.

### 5.2. Adjustment Scenario of GDP Growth Rate (Scenario 2)

The increase in GDP growth rate can increase the total GDP, thus, increasing steel consumption and promoting the realization of the carbon peak in advance. This scenario is to adjust the GDP growth rate and analyze the impact of different GDP growth rates on the carbon emissions of the steel industry from 2019 to 2050. On the basis of the baseline scenario, the system parameters are adjusted to set the parameter value in the GDP growth rate equation as 0.98 and when the parameter value is 0.98. At the same time, the $CO_2$ emission per unit of steel production is reduced by 10%, that is, from 1.92 tons to 1.728 tons, which is the impact of this scenario on the change of carbon emissions of steel industry in 2019–2050. Figure 4 shows the forecast chart of carbon emissions of iron and steel industry from 2019 to 2050 under this scenario.

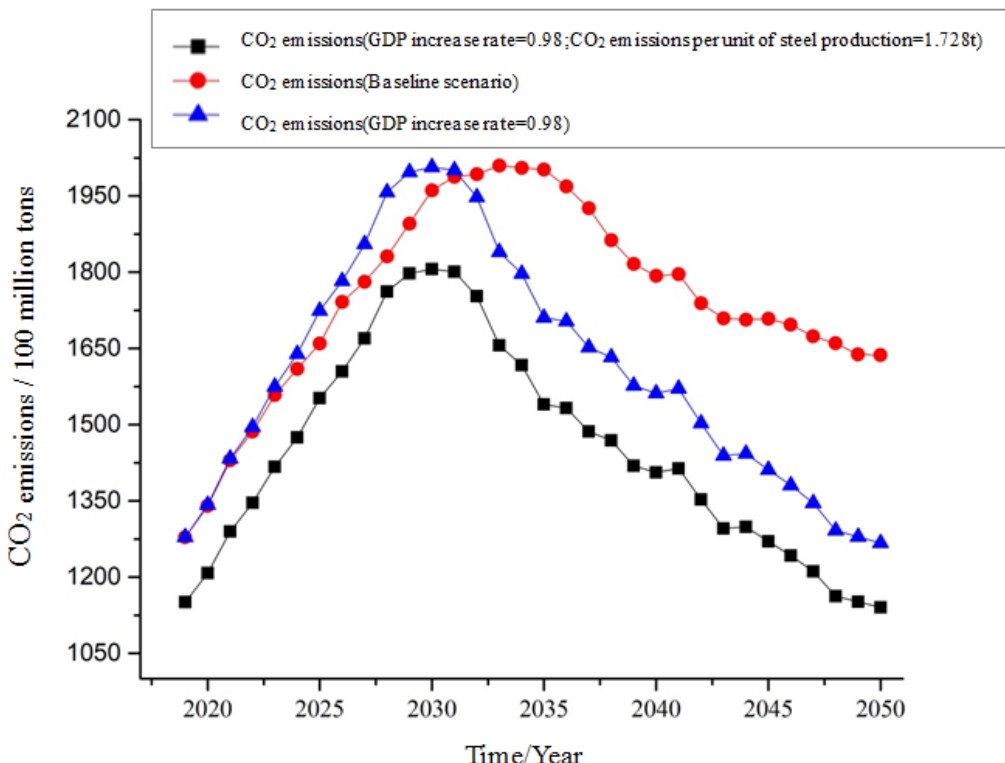

**Figure 4.** Forecast chart of $CO_2$ emissions from the steel industry (Scenario 2).

According to the prediction results of carbon emissions from the steel industry under the control of GDP growth rate in Figure 4, when the system parameter value is 0.98 and GDP growth rate is adjusted alone, 2030 is the earliest time for carbon to reach the peak, and the carbon peak is 20.64 billion tons, which decreases year by year thereafter. It can be inferred that carbon neutrality is expected to be achieved in 2060. At the same time, when we adjust the GDP growth rate and unit steel production of carbon emissions, that is, to improve the speed of economic development and enhance the level of steel production technology, carbon peak time is 2030; at this time, the carbon peak is 180.57 billion tons and, compared with the adjustment of GDP growth rate alone, is reduced by about 10%; compared with the baseline scenario, it is reduced by about 8.6%. Moreover, the carbon emissions of the steel industry after the peak are also significantly reduced. Compared with scenario 1, the carbon emissions after the peak are reduced by 2.26% on average, and the carbon neutrality goal is expected to be achieved before 2060.

### 5.3. Control Scenario of Steel Consumption in Construction Industry (Scenario 3)

By reducing the unit energy consumption required by the construction industry and accelerating the depreciation, the economic life of the building can be shortened, and then the steel consumption in the construction industry can be reduced. From the consumption end, the influence on the carbon emissions of the steel industry is analyzed. Based on scenario 1, the system variable parameters of the construction industry are adjusted to reduce the unit energy consumption by 20% and the service life of steel in the construction industry by 20%. That is, the system parameter value of unit energy consumption decreases from 1 to 0.8, and the system parameter value of steel service life decreases from 35 to 28. According to the system parameters of scenario 3, the development trend of carbon emissions in the steel industry from 2019 to 2050 is simulated, as shown in Figure 5. Compared with the baseline scenario, the predicted value of carbon emissions from 2019 to 2050 does not change significantly if the unit energy consumption in the construction industry and the service life of steel are reduced by 20%. As can be seen from the amount of steel in the construction industry (Figure 5), although the steel consumption in the

construction industry has decreased significantly after regulation, the proportion of steel consumption in the industry is smaller than that of the other three industries, resulting in no significant change in the carbon emissions of the steel industry.

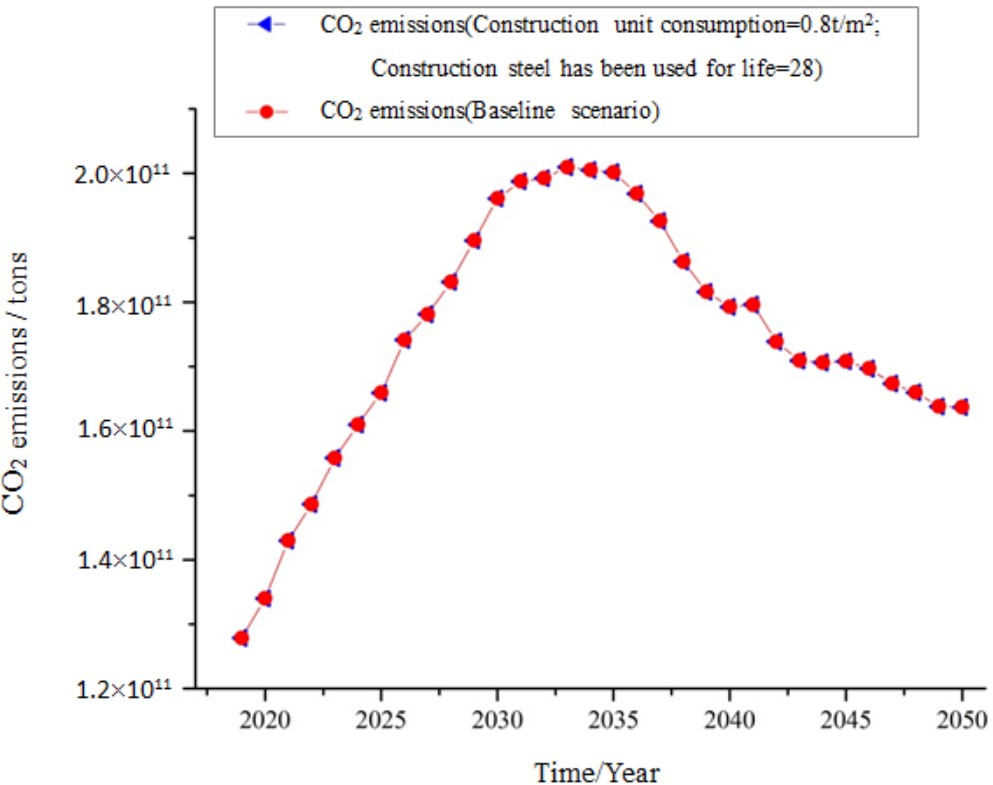

**Figure 5.** Forecast chart of $CO_2$ emissions from the steel industry (Scenario 3).

### 5.4. Control Scenario of Steel Consumption in Machinery Industry (Scenario 4)

Through accelerating the depreciation of machinery and equipment in the machinery industry to shorten its economic life and then reducing the steel consumption in the machinery industry, the influence of steel consumption change in the machinery industry on the carbon emissions of the steel industry is analyzed. Based on scenario 1, the system variable parameters of the machinery industry are adjusted to reduce the steel service life by 20%. That is, the system parameter value of the mechanical steel service life decreases from 19 to 15. According to the system parameters of scenario 2, the steel ownership of the machinery industry and the development trend of carbon emissions of the steel industry from 2019 to 2050 are simulated and predicted, as shown in Figure 6. As can be seen from Figure 6, reducing the service life of mechanical steel by 20% compared to the baseline scenario does not significantly change the projected carbon emissions for 2019–2050. Furthermore, through the analysis of the steel ownership of the machinery industry, it is found that although the steel consumption of the machinery industry has a certain decline after regulation, similar to the construction industry, its steel consumption change in the overall impact is not great, so the carbon emissions of the steel industry did not significantly reduce.

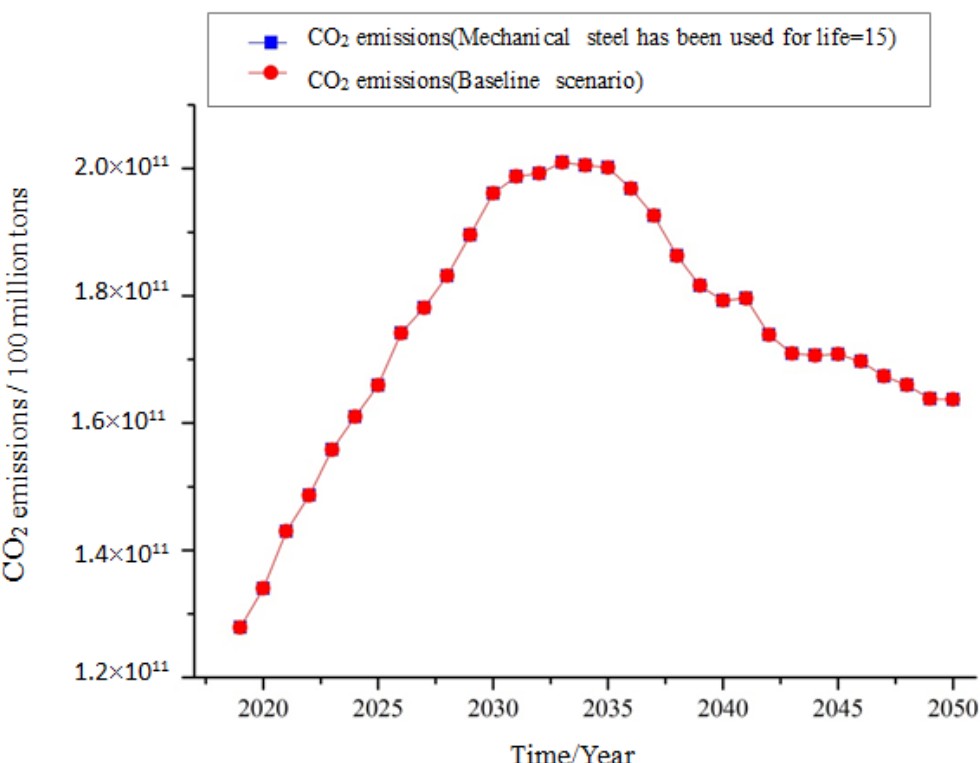

**Figure 6.** Forecast chart of $CO_2$ emissions from steel industry (Scenario 4).

*5.5. Scenario of Steel Consumption Control in Transportation Industry (Scenario 5)*

Through the accelerated depreciation of vehicles and ships in the transportation industry, their economic life is reduced so as to reduce the steel consumption in the transportation industry. The influence of the increase and decrease in steel consumption in the transportation industry on the carbon emissions of the steel industry is analyzed. We reduce the steel life of transportation vehicles by 20% based on the baseline scenario. That is, the system parameter value of the service life of steel in transportation vehicles decreases from 13.5 to 10.8. Accordingly, the development trend of carbon emissions in the steel industry from 2019 to 2050 is simulated and predicted, as shown in Figure 7. As can be seen from the figure, compared with the baseline scenario, if the service life of steel for the transportation vehicles is reduced by 20%, the predicted carbon emissions of the steel industry from 2019 to 2050 have a small fluctuation, and the average fluctuation range of carbon emissions is 0.22% compared with the baseline scenario. Although the regulation effect is not obvious, it still plays a certain role.

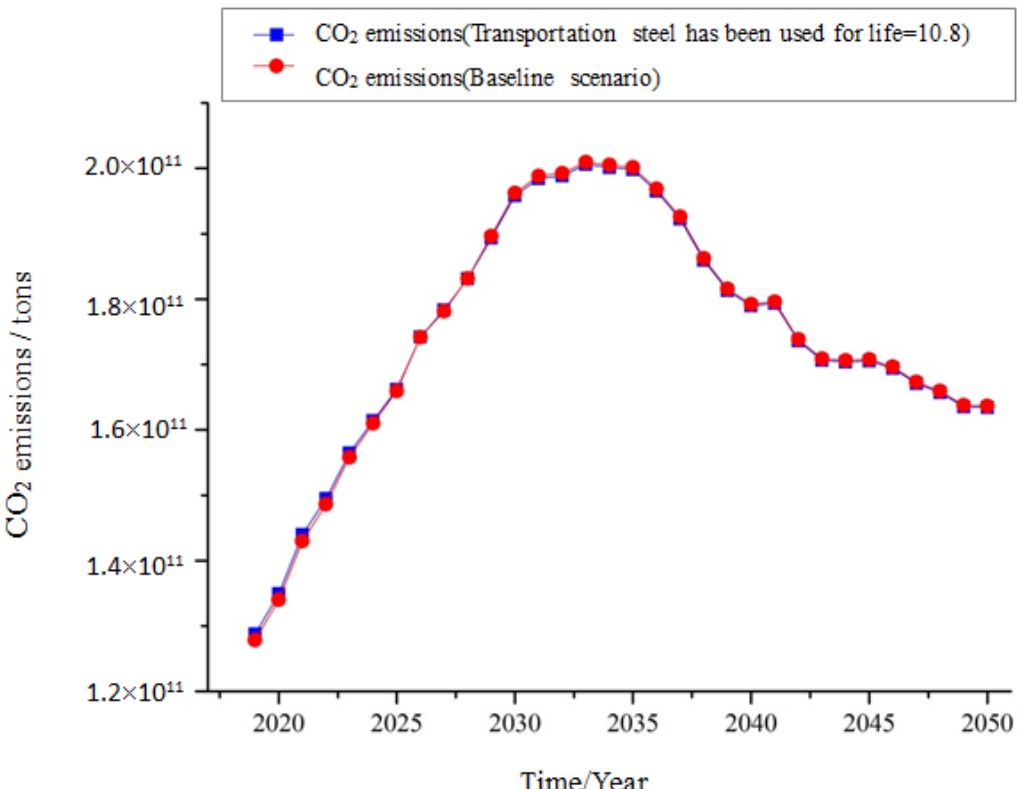

**Figure 7.** Forecast chart of $CO_2$ emissions from the steel industry (Scenario 5).

*5.6. Steel Consumption Control Scenario of Infrastructure Industry (Scenario 6)*

Infrastructure industry is involved in many areas, such as the highway, railway construction, thermal power and wind power. By enhancing the technical level to reduce highway railway construction steel consumption and alternative energy, we can reduce the use of serious ways of generating environmental pollution; increase nuclear power, as the future of new energy investment; shorten the service life of steel in the infrastructure industry at the same time and accelerate the equipment depreciation. Based on the baseline scenario, the service life is reduced by 20% and the original parameter value is changed from 25 to 20. In this scenario, the carbon emissions of steel industry from 2019 to 2050 are simulated and predicted. As shown in Figure 8, $CO_2$ emissions in scenario 6 show a clear downward trend, and the peak of carbon emissions is advanced from 2033 to 2031, which is closer to the target of carbon peak around 2030. At this time, the peak of carbon emissions is about 199.88 billion tons. According to scenario 6, namely, the regulation of steel consumption in the infrastructure industry has achieved obvious results. Compared with the first three industries, the reduction in steel consumption in the transportation industry can achieve a better emission reduction effect.

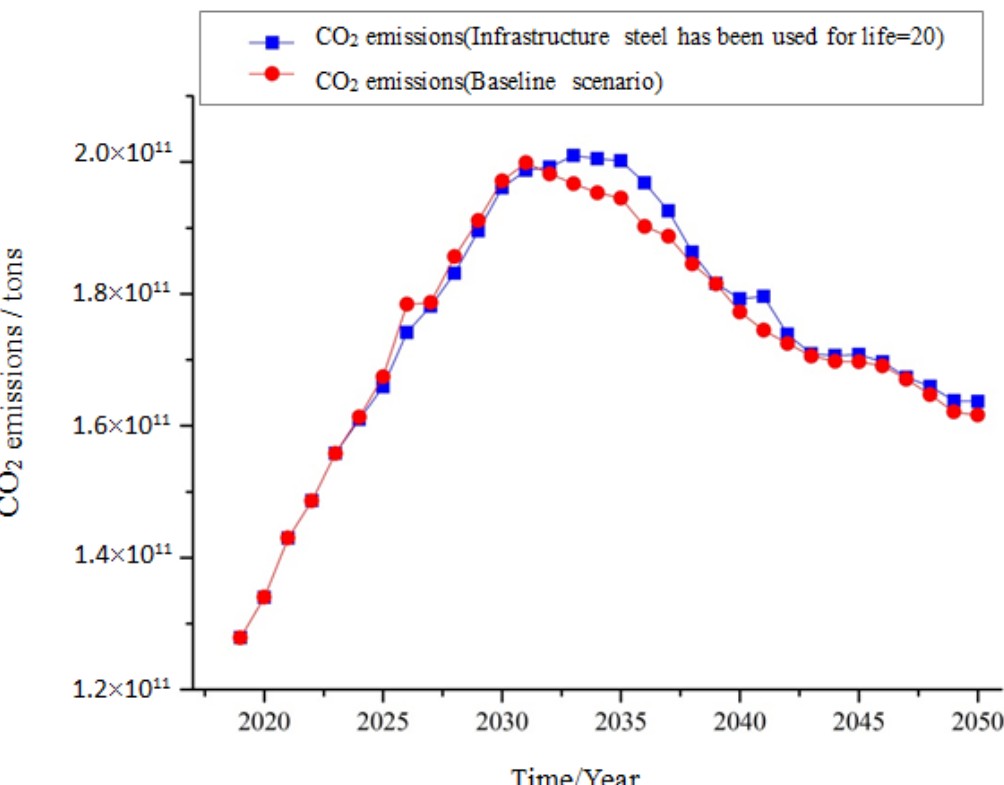

**Figure 8.** Forecast chart of CO2 emissions from the steel industry (Scenario 6).

*5.7. Comprehensive Regulation and Control of Steel Consumption in Four Industries (Scenario 7)*

In this scenario, the steel consumption in the construction industry, machinery industry, transportation industry and infrastructure industry are regulated, and the impact on the carbon emissions of the steel industry is analyzed. That is, in the baseline scenario, the system parameters of the four industries are adjusted at the same time so as to predict the change in trend of carbon emissions in the steel industry from 2019 to 2050. Figure 9 shows the carbon emission simulation prediction diagram under the comprehensive regulation of this scenario.

According to the carbon emission prediction results of comprehensive control in Figure 9, under the comprehensive control of steel consumption in the four industries, the carbon emission peak time of the steel industry is significantly earlier than that of the baseline scenario, and the carbon emission after the peak also decreases significantly. Compared with the baseline scenario, the peak time of carbon emissions in China's steel industry from 2019 to 2050 is four years earlier, and the peak time of carbon emissions is 2029, at which time the carbon peak is 198.66 billion tons, reaching the target of carbon peak before 2030. At the same time, after the peak time in 2029, Carbon emissions from China's steel industry also decrease by about 10.34% on average compared with the baseline scenario, with the decline rate increasing year by year except for a few years, and it is expected to achieve carbon neutrality by 2060.

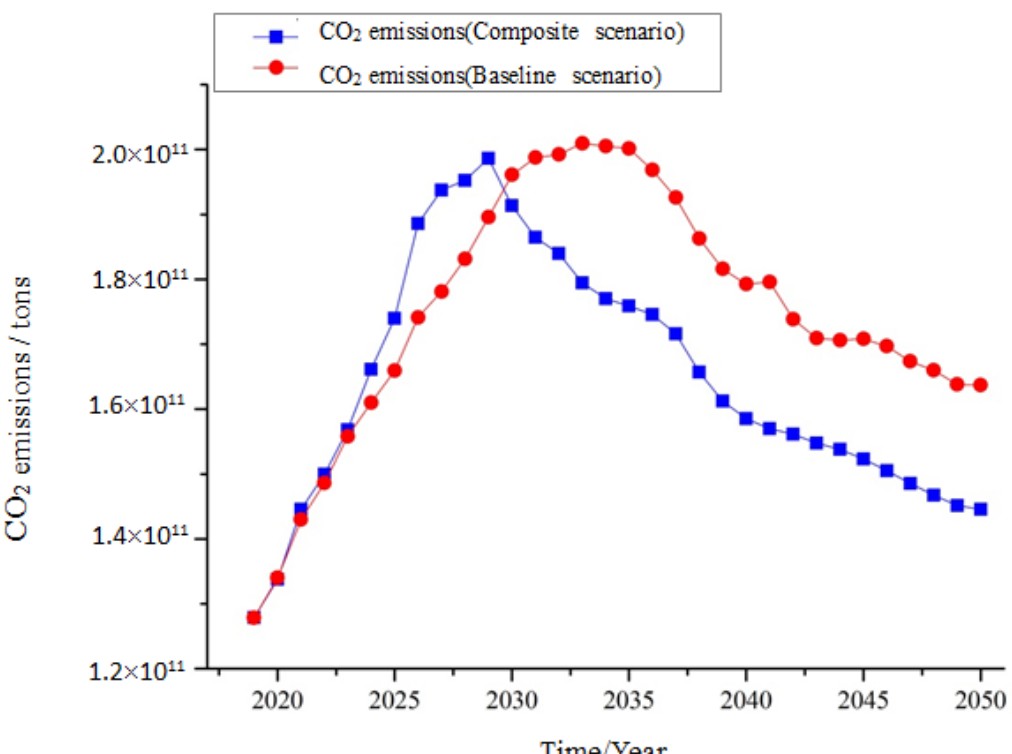

**Figure 9.** Forecast chart of $CO_2$ emissions from the steel industry (Scenario 7).

## 6. Conclusions

Based on the analysis of the interaction between steel consumption and steel industry carbon emissions and their influencing factors, this paper constructed a dynamic model of the steel consumption–carbon emissions system in China's steel industry and conducted a simulation analysis on the future carbon emission trend and carbon peak time. Seven different scenarios are used to predict the impact of carbon emissions in the steel industry and its carbon peaking time. It is expected to provide some reference for achieving the carbon peak and carbon neutrality targets in China's steel industry in the future.

(1)  After adjusting the GDP growth rate and carbon dioxide emission per unit of steel production, it is found that the carbon peak time of the steel industry is advanced to 2030, when the carbon peak is 1805.7, and the carbon emissions after the peak are greatly reduced; therefore, it is expected to reach the carbon neutrality goal before 2060. Therefore, in the future, China's steel industry can improve the level of production technology, reduce unit carbon emissions and reduce the overall carbon emissions of the steel industry, such as CCUS technology, carbon substitution technology, etc. At the same time, it can increase the GDP growth rate, increase the GDP and increase steel consumption, so as to advance the carbon peak time of the steel industry and strive to achieve the carbon peak in 2030 and carbon neutrality in 2060.

(2)  Found in the scenario simulation: The reduction in steel consumption in the construction and machinery sectors does not have a significant impact on carbon emissions from the steel industry, whereas the reduction in steel consumption in the transportation and infrastructure sectors has contributed to carbon reduction activities in the steel industry. The infrastructure industry, in particular, not only makes the iron and steel industry reduce the carbon emissions to have a certain margin, it also moves the prediction of carbon peak time closer to the aim of the carbon emission peak by 2030 from the baseline scenario early in 2033 to 2031. It shows that the reduction in steel consumption in the infrastructure industry has a good effect on carbon peak and carbon emission reduction in steel industry. In the future, China can actively build new green and low-carbon infrastructure and low-carbon transportation systems.

(3) When the four industries were regulated simultaneously, it was found that the predicted carbon peak time for the steel industry was advanced to 2029. At this time, the carbon peak will be 1986.6 million tons. The goal of achieving carbon peak by 2030 was accomplished, and carbon emissions will continue to decline after, with significant carbon reduction effects. As a result, future industries in China that consume steel can accelerate depreciation in the short term, and the life span of steel products and equipment can be shortened to promote the carbon peak time of the steel industry earlier. In the long term, technology can be improved and energy substitution can be implemented. Actively developing emission reduction pathways to cut steel consumption will reduce steel consumption and reduce carbon emissions in the steel industry to ensure that the "30–60 target" is achieved on schedule.

**Author Contributions:** Conceptualization, D.X.; methodology, D.X., E.L. and K.Y.; validation, D.X., E.L., W.D. and K.Y.; writing—original draft preparation, E.L. and K.Y.; writing—review and editing, D.X., E.L., W.D. and K.Y. All authors have read and agreed to the published version of the manuscript.

**Funding:** Supported by a project grant from National Natural Science Foundation of China (Grant No. 71764019), the National Social Science Foundation of China (Grant No. 19BGL187), the Planning Project of Philosophy and Social Science of Inner Mongolia (Grant No. 2021NDB082), Inner Mongolia Natural Science Foundation (Grant No. 2021MS07016 and 2022MS07012) and Basic Scientific Research Funds of Colleges and Universities Directly Under the Autonomous Region(Grant No. JY20220023 and JY20220055).

**Institutional Review Board Statement:** Not applicable.

**Informed Consent Statement:** Not applicable.

**Data Availability Statement:** Not applicable.

**Conflicts of Interest:** The authors declare no conflict of interest.

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
