# Peer review of "Consumption-Driven Carbon Emission Reduction Path and Simulation Research in Steel Industry: A Case Study of China"

_sustainability, doi:10.3390/su142013693_

Round 1
Reviewer 1 Report
There are many mistakes regarding the formats of writing:
1- At many parts of the paper you did not consider the rule of separation between words and sentences. For example page 2, line 3 of below, "by 2050;SEcond". Or Page 16, line 11, "steel industry;At the same".
2- Sometimes you have written some words with capital, like page 3, line 8,"that The increasing"!
Author Response
Dear reviewer:
Thank you for your decision and constructive comments on my manuscript. We have carefully considered the suggestion of Reviewer and make some changes. We have tried our best to improve and made some changes in the manuscript.
The red font that has been revised according to your comments in the revised manuscript. Revision notes, point-to-point, are given as follows:
Point 1: At many parts of the paper you did not consider the rule of separation between words and sentences. For example page 2, line 3 of below, "by 2050;SEcond". Or Page 16, line 11, "steel industry;At the same".
Response 1: We've proofread it carefully. For example page 5, line 25 after reformatting, "by 2050;SEcond", we have modified it to " by 2050. Second ". And page 19, line 13-14 of below after reformatting, "steel industry;At the same", we have modified it to " steel industry. At the same ".
Point 2: Sometimes you have written some words with capital, like page 3, line 8,"that The increasing"!
Response 2: We have revised them all. For example page 5, line 37 after reformatting,"that The increasing", we have modified it to " that the increasing ".
We apologize for the poor language of our manuscript. We worked on the manuscript for a long time and the repeated addition and removal of sentences and sections obviously led to poor readability. We have now worked on both language and readability and have also involved native English speakers for language corrections. We really hope that the flow and language level have been substantially improved.
Reviewer 2 Report
The article ''Carbon emission reduction path and simulation research based on consumption-driven Steel industry in China'' is presented for evaluation. There are multiple major and minor comments for the authors to improve the article in the following:
1. Title of the article should be attractive with respect to the novelty and scholarship point of view. This is very general statement style. Also, why there is ''*'' at the end of the title? please update accordingly.
2. The abstract start with a very long statement, which also contain some English language structure issue. Authors must avoid long statements, which result in vague outcome.
3. Abstract should start with the confined problem statement of the carbon emissions in steel industry. Authors should present some statistical figure which shows the danger level of the emissions, so that could be the base that why this study is needed.
4. Authors have mentioned the ''seven different development scenarios'' but didn't mentioned that how and why these are selected and what are the parameters / conditions were taken. Instead, authors directly mentioned about the outcome in the abstract.
5. Authors have mentioned ''china'' again and again in the abstract. In the start mentioning about the present study is focused on ''China'' once, should be fine.
6. Abstract needs complete revision in fact, authors should present their findings propely.
7. Authors have mentioned the word ''simulation based'' in the title, but I can not see any software / modelling / simulation software information, or any model description and how the simulation was carried out. The words in the titles should be chosen with great care.
8. It seems that authors have not proof read the article before submission that it contain many typos and English language errors. e.g. ''56. 71 percent'', there is gap between 56. and 71 and also ''percent'' should be written as ''%''. Please go through the whole article and check for minor typos etc. and authors must highlight in the revision version for all such mistakes.
9. Figure 1: text is not visible, instead of so complex mechanism, authors should at least draw it at a level that it should be readable.
10. In Table 1 GDP column, authors should concise it to more decimal by K or Million'' and to change the currency from ''yuan'' to ''USD'' and / or to compare it with the already present literature.
11. Many Tables have same data which is in figures as well, e.g. table 2 and figure 2 have same data or part of same data. This is a research paper, not a report, where authors may require to increase the length. Authors should have enough information about the presentation of the data in the research paper and only for the novel and specific results should be presented.
12. Authors should highlight and mention in the revised version regarding the deletion of doubling and/or overlapped data.
13. Almost all figures and tables data is overlapping. please remove and also to improve the data analysis. Usually, repetition of same type of graphs / figures is not required. Instead more data analyses may give better understanding and readership of the paper.
14. Conclusion section is very lengthy and it require a lot of improvements.
15. There are only 19 references and few are very old. Authors are advised to double the references and also to add few references of this journal for better relevancy to this journal and good readership.
Author Response
Dear reviewer:
Thank you for your decision and constructive comments on my manuscript. We have carefully considered the suggestion of Reviewer and make some changes. We have tried our best to improve and made some changes in the manuscript.
The red font that has been revised according to your comments in the revised manuscript. Revision notes, point-to-point, are given as follows:
Point 1: Title of the article should be attractive with respect to the novelty and scholarship point of view. This is very general statement style. Also, why there is ''*'' at the end of the title? please update accordingly.
Response 1: Thank you for the title suggested. After our serious discussion, we decided to change the title to " Consumption-driven carbon emission reduction path and simulation research in Steel industry: A case study of China ". We hope to be more attractive in terms of novelty and scholarship point. And the ''*'' at the end of the title is because a footnote about the authors has been added after the title. We have put the footnote after the author's name.
Point 2: The abstract start with a very long statement, which also contain some English language structure issue. Authors must avoid long statements, which result in vague outcome.
Response 2: We apologize for some language structure problems in the translation of the abstract. We have modified the beginning of the abstract to a shorter sentence. In fact, the abstract was almost changed to shorter sentences. Now our abstract begins more simply and clearly.
Point 3: Abstract should start with the confined problem statement of the carbon emissions in steel industry. Authors should present some statistical figure which shows the danger level of the emissions, so that could be the base that why this study is needed.
Response 3: We have modified the abstract to begin with the confined issue of carbon emissions in the steel industry. And some statistics figure about carbon emissions are provided. As you can see, our abstract starts with the proportion of carbon emissions from China's steel industry to the world's steel industry. It can visually illustrate the importance of carbon reduction in China's steel industry.
Point 4: Authors have mentioned the ''seven different development scenarios'' but didn't mentioned that how and why these are selected and what are the parameters / conditions were taken. Instead, authors directly mentioned about the outcome in the abstract.
Response 4: In this paper, seven different development scenarios are set up to adjust the GDP growth rate, the system parameter values of each element in the construction, machinery, transportation and infrastructure industries, as well as the baseline scenario and the integrated regulation. The paper analyzes and predicts how the carbon emissions of the steel industry will be affected in the future in different development scenarios. But we have removed " seven different development scenarios " from the abstract due to word limit. And increased the length of our findings.
Point 5: Authors have mentioned ''china'' again and again in the abstract. In the start mentioning about the present study is focused on ''China'' once, should be fine.
Response 5: Thank you for your suggestion. We have included a presentation on the carbon emissions of the China’s steel industry at the beginning of the introduction. You can see it in the revised manuscript.
Point 6: Abstract needs complete revision in fact, authors should present their findings propely.
Response 6: We have completely revised the abstract and accurately presented our findings. For example, by adjusting the GDP growth rate and CO2 emissions per unit of steel production, the carbon peak in the steel industry is advanced to 2030, and the carbon emissions after the peak are significantly reduced. And the reduction in steel consumption in the construction and machinery sectors does not have a significant impact on carbon emissions from the steel industry and so on.
Point 7: Authors have mentioned the word ''simulation based'' in the title, but I can not see any software / modelling / simulation software information, or any model description and how the simulation was carried out. The words in the titles should be chosen with great care.
Response 7: We have included in our literature review about system dynamics simulations. And we have added steps about simulation runs and informations about software in the second part. You can view the second part of the revised manuscript.
Point 8: It seems that authors have not proof read the article before submission that it contain many typos and English language errors. e.g. ''56. 71 percent'', there is gap between 56. and 71 and also ''percent'' should be written as ''%''. Please go through the whole article and check for minor typos etc. and authors must highlight in the revision version for all such mistakes.
Response 8: We apologize for the language errors in our manuscript. We worked on the manuscript for a long time, and repeatedly adding and removing sentences and chapters apparently led to poor readability. And there were some typos in the translation of the manuscript. Now we have worked on both language and readability, and have involved native English speakers in the language correction. We really hope to make substantial improvements in both fluency and language level. We have revised these typos and English language errors and tracked the changes in a reformatted manuscript. For example "56. 71 percent ", we have modified it to " 56. 71% ".
Point 9: Figure 1: text is not visible, instead of so complex mechanism, authors should at least draw it at a level that it should be readable.
Response 9: Thank you very much for your suggestion. We have re-uploaded more visible and clearer figure 1.
Point 10: In Table 1 GDP column, authors should concise it to more decimal by K or Million'' and to change the currency from ''yuan'' to ''USD'' and / or to compare it with the already present literature.
Response 10: We have modified the "348518 " to "34.8518" in Table 1. And the same changes were made to other data. We have modified the units of GDP to "1000 billion RMB ". Since we are studying China's GDP, we use the "RMB". We hope these revisions will give readers a better experience.
Point 11: Many Tables have same data which is in figures as well, e.g. table 2 and figure 2 have same data or part of same data. This is a research paper, not a report, where authors may require to increase the length. Authors should have enough information about the presentation of the data in the research paper and only for the novel and specific results should be presented.
Response 11: We're so sorry. We thought the data needed to be listed as well, so we put it in tables. We describe and analyze the data in a more descriptive way. And only novel and specific results are presented. For example, in scenario 1, more representation and analysis was done for Figure 2.
Point 12: Authors should highlight and mention in the revised version regarding the deletion of doubling and/or overlapped data.
Response 12: The problem of overlapping data has been highlighted and mentioned in the revised manuscript.
Point 13: Almost all figures and tables data is overlapping. please remove and also to improve the data analysis. Usually, repetition of same type of graphs / figures is not required. Instead more data analyses may give better understanding and readership of the paper.
Response 13: We have removed tables or figures that have overlapping data. Because there are scenario simulations in which the data have barely changed compared to the baseline scenario. So this part will not have much data analysis as far as the graphs are concerned. And the graphs in the other scenarios were analyzed for data.
Point 14: Conclusion section is very lengthy and it require a lot of improvements.
Response 14: We have rewritten the conclusion section. Some lengthy sentences have been removed to make it as clear and concise as possible. For more details, please see the conclusion section in the revised manuscript.
Point 15: There are only 19 references and few are very old. Authors are advised to double the references and also to add few references of this journal for better relevancy to this journal and good readership.
Response 15: Due to the addition of the literature review on steel consumption and system dynamics, we have increased the number of references to 41 and updated them to the latest. In addition to this, some references from this journal have been added.
Reviewer 3 Report
The article addresses important and current problems related to carbon emission reduction path and simulation research based on consumption-driven Steel industry in China.
Please adapt the manuscript to the journal's requirements. The current version of the manuscript requires improvement, e.g. no spaces before square brackets, the way of saving publications in References and others.
The literature review is incomplete. There are no references in the manuscript to the latest scientific publications on similar topics (also from 2022). Moreover, there are far too few references to other scientific publications.
The Conclusions section should be completed (also taking into account the extended literature review). Please indicate the potential use of the research results.
Author Response
Dear reviewer:
Thank you for your decision and constructive comments on my manuscript. We have carefully considered the suggestion of Reviewer and make some changes. We have tried our best to improve and made some changes in the manuscript.
The red font that has been revised according to your comments in the revised manuscript. Revision notes, point-to-point, are given as follows:
Point 1: Please adapt the manuscript to the journal's requirements. The current version of the manuscript requires improvement, e.g. no spaces before square brackets, the way of saving publications in References and others.
Response 1: Thank you very much for your suggestion. We have revised the manuscript in accordance with the requirements of the journal. Spaces have been added before all square brackets. You can see it in the introduction and literature review in our revised manuscript. And the format of the references has been revised accordingly.
Point 2: The literature review is incomplete. There are no references in the manuscript to the latest scientific publications on similar topics (also from 2022). Moreover, there are far too few references to other scientific publications.
Response 2: Due to the addition of the literature review on steel consumption and system dynamics, we have increased the number of references to 41 and updated them to the latest.
Point 3: The Conclusions section should be completed (also taking into account the extended literature review). Please indicate the potential use of the research results.
Response 3: We have rewritten the conclusion section. The potential use of the research results was also pointed out to provide some reference for the China’s steel industry to achieve the carbon peak and carbon neutrality target in the future. For more details, please see the conclusion section in the revised manuscript.
Round 2
Reviewer 2 Report
Accept
Reviewer 3 Report
The revised manuscript may be accept in present form. However, it must be remembered that all minor errors must be corrected eg in "CO2" - "2" should be subscript - the same error is in three different places in the manuscript.